# LEARNING WHAT TO LEARN IN A NEURAL PROGRAM

## ABSTRACT

Learning programs with neural networks is a challenging task, addressed by a long line of existing work. It is difficult to learn neural networks which will generalize to problem instances that are much larger than those used during training. Furthermore, even when the learned neural program empirically works on all test inputs, we cannot verify that it will work on every possible input. Recent work has shown that it is possible to address these issues by using recursion in the Neural Programmer-Interpreter, but this technique requires a *verification set* which is difficult to construct without knowledge of the internals of the *oracle* used to generate training data. In this work, we show how to automatically build such a verification set, which can also be directly used for training. By interactively querying an oracle, we can construct this set with minimal additional knowledge about the oracle. We empirically demonstrate that our method allows automated learning and verification of a recursive NPI program with provably perfect generalization.

## 1 INTRODUCTION

In recent years, the use of larger datasets and bigger models with greater representational capacity has led to significant advances in many applications such as object recognition in images and machine translation. Inspired by this progress, many researchers have used neural networks for program induction, especially through the development of novel neural network architectures which feature components such as a variable-size memory (Sukhbaatar et al., 2015; Kurach et al., 2016; Graves et al., 2014; Joulin & Mikolov, 2015b). Compared to baseline approaches like recurrent neural networks and LSTMs, these architectures are able to learn more effectively from input-output examples on tasks such as addition, sorting, and permutation of sequences, as measured by their empirical accuracy on a held-out test set of input-output examples.

For program induction, there typically exists a parsimonious underlying program to solve the problem which has been specified using input-output example pairs. However, as the space of all programs is extremely large, it is often difficult for neural networks to learn the correct underlying program just using input-output examples. Indeed, many of the prior works report that the learned neural network empirically fails to generalize to significantly larger inputs than those in the training data, which indicates that the neural network has learned spurious dependencies on irrelevant idiosyncrasies of the training data (such as length of each training example). These failures occur despite the use of approaches like curriculum learning (Bengio et al., 2009), where the training data initially consists of easy examples and gradually becomes more complicated as training progresses. Furthermore, the neural network architectures may be very sensitive to hyperparameter settings, with the best generalization results only achieved in a fraction of a percent of the hyperparameter space (Kaiser & Sutskever, 2016).

Even when a learned neural program exhibits empirical generalization to arbitrarily complicated inputs, a more fundamental issue remains: while the learned neural program may empirically produce the correct result on every input example attempted, we cannot show that the neural network will operate correctly on every other possible input. Without actually running the neural network on a given input, it is quite difficult to predict or otherwise characterize its behavior on that single input, let alone a large (often infinitely large) class of inputs. Nevertheless, we would like a *proof of correctness* that the learned neural network has learned the right underlying program, and will therefore operate correctly on any input.

Prior work by Cai et al. (2017) addresses the problem of proving correctness of a learned neural network program by introducing recursion to the Neural Programmer-Interpreter (NPI) architec-

ture (Reed & de Freitas, 2016). Unlike most other architectures designed for learning programs and solving algorithmic tasks, the Neural Programmer-Interpreter emphasizes the compositional nature of programs, solving a problem through functions which can call other functions. Both the original work by Reed & de Freitas (2016) and the later work by Cai et al. (2017) train the architecture not with just input-output pairs, but with execution traces which describe in detail the role of each function in solving a given input problem. Cai et al. (2017) ensured that these traces are *recursive*: each function only takes a finite, bounded number of actions. To solve problems where the number of actions needed grows with the size of the problem, the function calls itself to perform the necessary repetition. This property not only led to empirically better generalization compared to the earlier work, but enabled the authors to formally verify that the learned neural programs would generalize to any input.

**Problem statement and proposed approach.**    For any of the prior neural program architectures, the goal is to train a neural network to duplicate the behavior of an *oracle* which can solve any instance of the problem at hand. In this work, we seek to answer the question of how to generate a suitable training set for learning such a neural program architecture so that it can successfully duplicate the behavior of the oracle. In other words, what is the set of input examples for which we should demand labels from the oracle? To our knowledge, prior work does not explicitly address this question: the usual practice has been to use a "large enough" training set in the hope that the resulting learned neural program will be good enough.

Indeed, prior work usually assumes that a fixed set of data is available, and treats the task as learning something from this fixed set. Our goal is to learn the true underlying program—if we are only given a fixed set of data, it could easily be that this data does not demonstrate all of the behaviors of the latent program (see Section B for more discussion). To overcome this issue, the setting in this paper is closer to active learning, where we assume that we can query an oracle with previously unlabeled data points to obtain more labels. Many past works, through their combination of curriculum learning Bengio et al. (2009), and dynamic generation of new problem instances then solving them with an oracle to obtain each mini-batch of training data, also use a setting similar to active learning (see Section C).

We work with recursive NPI oracles from Cai et al. (2017), as they provide a detailed execution trace that describes how to solve a problem in terms of smaller functions. We iteratively explore all possible behaviors of the oracle in a breadth-first manner, and the bounded nature of the recursive oracle ensures that our procedure reaches a fixed point in finite time. This method automatically identifies a sufficient training set for fully learning the behavior of the oracle, and perfect accuracy on this training set provides a proof that the learned neural program will generalize to any input.

Furthermore, once we have a complete training set which fully describes the behavior of the oracle, we can identify and remove redundant information in each trace of the set, to significantly reduce the set's size and enable faster training. The minimized set also allows for provably perfect duplication of the oracle.

**Contributions of this work.**    We make the following contributions:

- We provide an algorithm for automatically generating a sufficiently large and diverse NPI training set which, by construction, allows us to exactly mimic the oracle's behavior on *any* valid input.

- We provide a formal proof of correctness that this algorithm, for a bounded-length NPI oracle adhering to some mild conditions, will produce execution traces which cover all of the possible behaviors of the oracle.

- We also demonstrate a method for removing *irrelevant* observations in each trace, which allows us to significantly reduce the size of a NPI training set needed to mimic an oracle through deduplication of the simplified traces.

- As a consequence of the above, we automate the process of showing *provably perfect generalization* (Cai et al., 2017) for a learned recursive NPI program; we automatically generate a training set, such that achieving perfect accuracy on this set guarantees perfect generalization.

We empirically validate our methods on the addition, bubblesort, and topological sort tasks from Cai et al. (2017). Our experimental results show that, with only black-box access to the oracle, we can automatically generate a small training set. A neural program learned from this training set empirically generalizes to all attempted inputs; furthermore, we show that a neural program which achieves perfect accuracy on the training set is guaranteed to give correct results on *any* input.

As shown in Section B of the appendix, manually creating a suitable training set can take trial and error. Manually creating a verification set for provable generalization (Cai et al., 2017) requires careful reasoning about the internal mechanisms of the oracle. In contrast, our approach provides a complete and automated solution to the problem of learning a neural program with provably perfect generalization, with only black-box access to the oracle, for our target domain of recursive NPI oracles.

## 2 BACKGROUND: NEURAL PROGRAMMER-INTERPRETER

In this section, we review the Neural Programmer-Interpreter architecture by Reed & de Freitas (2016), with an emphasis on the aspects that are most salient for our contributions.

The Neural Programmer-Interpreter architecture consists of three components: a *core module* shared across all tasks, learned *function embeddings* which direct the core module, and *domain-specific encoders* which summarize the *environment* into a fixed-size representation and provides it as an input to the core module.

The core module is recurrent and implemented as an LSTM (Hochreiter & Schmidhuber, 1997). At each step, it receives the embeddings for the current function being executed ($p$) and its arguments ($a$), the domain-specific encoder's ($f_{enc}$) *observation* of the environment ($e_t$), and the previous hidden state ($h_{t-1}$); it produces the next function to run (and arguments for the function) using content-based addressing ($k_t$), whether to return control to the caller function in the next step ($r_t$; $0 \leq r_t \leq 1$), and the next hidden state ($h_t$). More formally, we write

$$h_t = LSTM(f_{enc}(e_t), p, a, h_{t-1})$$
$$r_t = f_{end}(h_t), p' = f_{prog}(h_t), a' = f_{arg}(h_t)$$

The LSTM applies a small neural network, e.g. with two hidden layers, to merge the inputs for the current timestep.

If $p'$ is a *primitive* function, we follow its (hardcoded) definition to manipulate the environment. Otherwise, if $p'$ is *not* a primitive function, we suspend the execution of the current function $p$ and transfer control to $p'$: the current hidden state $h_t$ is set aside, and in the next step of execution, the NPI core receives 0 as the previous hidden state and $p'$ as the current function being executed. When $p'$ returns control to the caller (when $r_t > 0.5$), we restore the set-aside hidden state and the program embedding for the caller $p$ in the next step of execution.

Conceptually, we can view the Neural Programmer-Interpreter as producing a long sequence of primitive function calls, each of which manipulate the environment in some way, until the environment reaches a desired state. The observations (produced by $f_{enc}$ summarizing an environment) inform the core module which functions to invoke.

**Example task: bubblesort.** For the purpose of exposition, we will examine the *bubblesort* task from Reed & de Freitas (2016) and Cai et al. (2017) throughout the paper.

In this task, the environment is a one-hot encoded *scratch pad* $Q \in \mathbb{R}^{N \times K}$ where $N$ is the array length and $K$ is the one-hot encoding dimension (number of possibilities for each array entry). We will sort decimal digits, so $K = 10$. The environment also contains three pointers called p1, p2, and p3. We initialize the environment with the array we wish to sort, and the pointers at location 0. We will sort arbitrarily large arrays, so the environment can also have unbounded size.

$f_{enc}$ encodes the values at p1 and p2, and whether p3 is within bounds or beyond the length of the array. Therefore, the observation is a fixed-size tuple of these three values, whereas the environment can have arbitrary size depending on the length of the array.

Figure 1 shows an example execution trace. The primitive function PTR moves a pointer by one location left or right, as specified by the arguments; SWAP switches the array values pointed to by

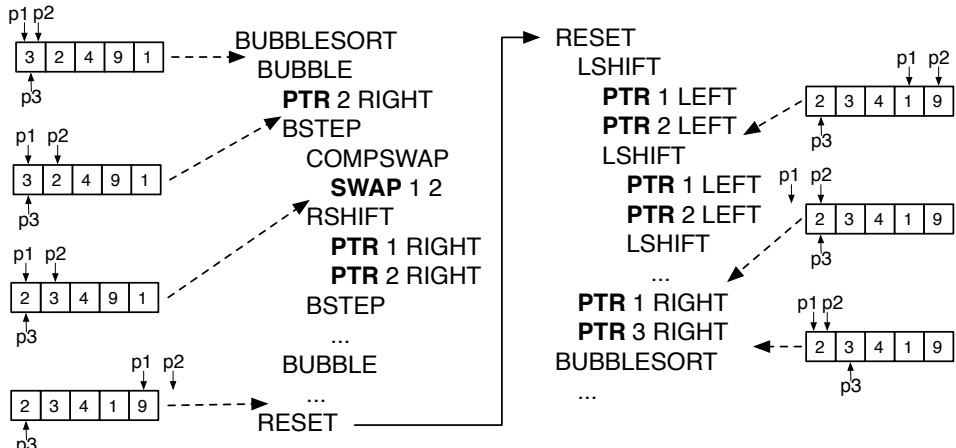

Figure 1: A partial execution trace for the bubblesort task. Ellipses denote elided portions of the trace. The dashed arrows show the environment at various points of execution. Primitive functions are bold.

| | Input | | Output | |
|---|---|---|---|---|
| Function | Conditions on $\bigcup_{n \in \mathbb{N}} \mathcal{O}^n$ | | Function | $r_t$ |
| BSTEP | $n = 1 \wedge o_1[p2] = \texttt{end}$ | | NOP | 1 |
| | $n = 1 \wedge o_1[p2] \neq \texttt{end}$ | | COMPSWAP | 0 |
| | $n = 2$ | | RSHIFT | 0 |
| | $n = 3$ | | BSTEP | 1 |
| COMPSWAP | $n = 1 \wedge o_1[p1] \leq o_1[p2]$ | | NOP | 1 |
| | $n = 1 \wedge o_1[p2] > o_1[p2]$ | | SWAP 1 2 | 0 |

Table 1: Tabular representation of Oracle for BSTEP and COMPSWAP.

two pointers; NOP does nothing. The (non-primitive) function BUBBLESORT performs one sorting pass through the array. BUBBLE performs one sweep left to right, BSTEP performs one step in this sweep, RESET returns the pointers back to their original locations, and COMPSWAP conditionally swaps two elements.

**Execution traces and the oracle.** In order to train the NPI architecture, we use an *oracle* to obtain an *execution trace* that describes the behavior of each function. The oracle can be described by a function[1]

$$\texttt{Oracle} : F \times A \times \left( \bigcup_{n \in \mathbb{N}} \mathcal{O}^n \right) \rightarrow (F \cup P) \times A \times \{0, 1\} \tag{1}$$

where $F$ is the set of non-primitive functions, $P$ is the set of primitive functions, $A$ is the set of all arguments, $\mathcal{O}$ is the set of all observations (i.e., the range of $f_{enc}(\cdot)$), and $\bigcup_{n \in \mathbb{N}} \mathcal{O}^n$ is the set of all sequences of observations. The last part of the output $\{0, 1\}$ corresponds to $r_t$: whether to return control to the caller function.

Table 1 shows a subset of Oracle for the bubblesort task, for the BSTEP and COMPSWAP functions. Each row represents a possible output of Oracle, and columns 1 and 2 show the conditions for when it would have that output.

Oracle is not defined over all of $F \times A \times \left( \bigcup_{n \in \mathbb{N}} \mathcal{O}^n \right)$ because not all combinations of $F$ and $A$ are allowed, and also because most functions will not execute for an arbitrary number of steps. For example, in Table 1, BSTEP executes for at most three steps ($1 \leq n \leq 3$) and COMPSWAP only for one step.

While past work using NPI did not explicitly define an oracle in this way, we would like to emphasize that the training data used by past work nevertheless needs to have originated from a generative

---

[1] Without loss of generality, we consider each function to only take one argument.

process which can be described by such a function. Otherwise, it would not be feasible for the NPI model to learn the behavior of the oracle accurately, because there is not enough information provided to the NPI core at inference time to unambiguously reproduce the oracle's response.

## 3 CREATING A TRAINING SET FROM AN NPI ORACLE

Now that we have formally defined the Neural Programmer-Interpreter and the oracle, we will now discuss how to automatically generate a training set of execution traces by querying the oracle. In summary, we build trees describing all possible behaviors of the oracle. We expand them breadth-first in an iterative manner, as we learn about the oracle's response to each observation sequence. Our procedure reaches a fixed point due to the boundedness of the oracle. After completion, traversals of the trees form a complete training and verification set for the oracle.

### 3.1 ENUMERATING THE ORACLE'S BEHAVIOR

So that we can train the NPI core LSTM to duplicate the oracle's behavior on any input, we would like to record the oracle's response to all possible combinations of functions, arguments, and observation sequences that may arise during execution on a valid problem instance. However, it is untenable to query the oracle for all elements in $F \times A \times (\bigcup_{n \in \mathbb{N}} \mathcal{O}^n)$, with an immediate obstacle being that this set is infinitely large. Even if we know that all functions in the oracle only execute for a bounded number of steps, we may not know the precise bound.

Instead, we will assume that we know the set of possible *initial observations* and the *entry function* that begins every execution trace. In our bubblesort example, the entry function is BUBBLESORT (as seen in Figure 1) and the set of initial observations is $\{(\mathsf{p1} = i, \mathsf{p2} = i, \mathsf{p3}\ \mathsf{in}\ \mathsf{bounds} = 1) : i \in \{0, 1, \cdots, 9\}\}$ since $\mathsf{p1}$ and $\mathsf{p2}$ initially point to the same position in the array.

The initial observations and the entry function form a subset of $F \times A \times (\bigcup_{n \in \mathbb{N}} \mathcal{O}^n)$, which we shall call $Q_0$. By querying the oracle on $Q_0$, we can obtain $Q_1 \subset F \times A \times (\bigcup_{n \in \mathbb{N}} \mathcal{O}^n)$; query the oracle on $Q_1$ to create $Q_2$; and so on, until we observe no growth in $Q$ (i.e. $\bigcup_{i \leq n-1} Q_i = \bigcup_{i \leq n} Q_i$). Then by taking $\bigcup_i Q_i$ and the corresponding responses of the oracle, we obtain the training data needed to clone the oracle's behavior.

More specifically, let us denote an arbitrary element of $(f, a, [o_0, \cdots, o_i]) \in Q_i$. We can then query the oracle on this element to obtain $(f', a', r_i) = \mathtt{Oracle}(p, a, [o_0, \cdots, o_i])$. If $f'$ is a non-primitive function, then $(f', a', [o_i]) \in Q_{i+1}$. If $f'$ is a primitive function, we compute $\hat{f}'(o_i) = O_{i+1} \subset \mathcal{O}$, the set of observations we can obtain after invoking $f'$, and add $(f, a, [o_0, \cdots, o_i, o_{i+1}])$ to $Q_{i+1}$ for each $o_{i+1} \in O_{i+1}$. The next section discusses $\hat{f}'$ in greater depth.

Please refer to Algorithm 1 in the appendix for a full description of the procedure and proof of its correctness and termination.

**Bubblesort example.** In bubblesort, an element of $Q_0$ is $(\mathtt{BUBBLESORT}, (), [(\mathsf{p1} = 3, \mathsf{p2} = 3, \mathsf{p3}\ \mathsf{in}\ \mathsf{bounds} = 1)])$. As shown in line 2 of Figure 1, $\mathtt{Oracle}(\mathtt{BUBBLESORT}, (), [(\mathsf{p1} = 3, \mathsf{p2} = 3, \mathsf{p3}\ \mathsf{in}\ \mathsf{bounds} = 1)]) = (\mathtt{BUBBLE}, (), 0)$. Therefore, we can add $(\mathtt{BUBBLE}, (), [(\mathsf{p1} = 3, \mathsf{p2} = 3, \mathsf{p3}\ \mathsf{in}\ \mathsf{bounds} = 1)])$ to $Q_1$.

### 3.2 EXECUTING OVER OBSERVATIONS INSTEAD OF STATES

Normally, we execute the instructions from the oracle (or from an NPI core after it has been trained) on a concrete environment. Each primitive function call directly transforms a given environment to a different environment. A non-primitive function also indirectly transforms a given environment to a particular different environment (a single one, assuming determinism of the oracle) through making a series of primitive function calls.

However, the set of valid environments can be infinitely large, because the initial environment must encode the input problem exactly and there is an infinite number of input problems (e.g. sorting of arbitrarily long arrays). Furthermore, sequences of observations determine the oracle's behavior, not the environments themselves. Therefore, we base our analysis on observations, even though

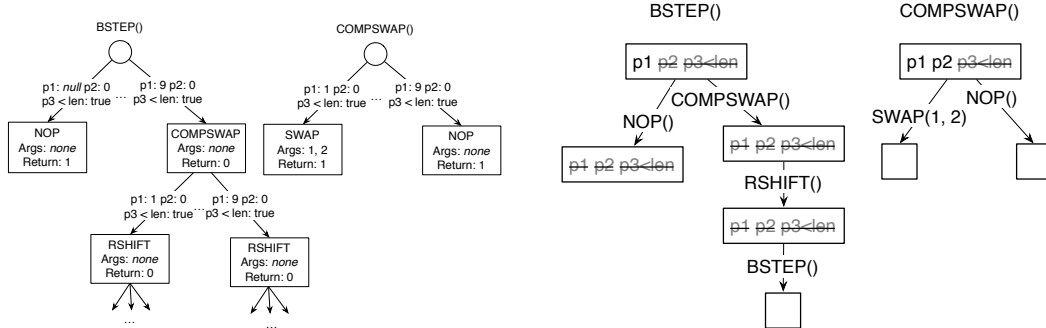

(a) $\text{Oracle}_{BSTEP,()}$ and $\text{Oracle}_{COMPSWAP,()}$ generated by Algorithm 1 for the bubblesort task.

(b) The crossed-out text corresponds to the observation dimensions which are irrelevant for deciding the next action.

Figure 2: The main outputs of our proposed methods, in the form of trees. See Sections 3.3 and 4 for more details.

environments represent the actual state of execution. This is similar to *abstract interpretation* (Cousot & Cousot, 1977). We replace the environments (in bubblesort: the entire array being sorted) which are *concrete* states with observations (bubblesort: value at p1, value at p2, and whether p3 is within bounds) as *abstract* states.

For this, we need a correspondence to primitive functions which operate over observations instead of environments. Consider a primitive function $f : A \times E \to E$, which operates on some environment $e_i$ and transforms it into a different environment $e_j$. We will now define $\hat{f} : A \times \mathcal{O} \to 2^{\mathcal{O}}$ as the following:

$$\hat{f}(a, o) = \{o' \in \mathcal{O} \mid \exists e_i, e_j \in E.\ f(a, e_i) = e_j \wedge f_{enc}(e_j) = o'\}.$$

Informally, $\hat{f}(a, o)$ gives the set of all observations $o'$ we could obtain if we run the primitive function $f$ with argument $a$ from all possible environments $e$ where $f_{enc}(e) = o$. We assume that $\hat{f}$ is given for each primitive function $f$.

**Bubblesort example.** We have two primitive functions: PTR moves a pointer to the left or right, and SWAP swaps the value under two pointers. Then we have

$$\hat{f}_{\text{PTR}}((p, \text{LEFT or RIGHT}), (\text{p1} = v_1, \text{p2} = v_2, \text{p3 in bounds} = v_3)) =$$

$$\begin{cases} \{(\text{p1} = i, \text{p2} = v_2, \text{p3 in bounds} = v_3) \mid i \in \{0, \cdots, 9\}\} & \text{if } p = 1 \\ \{(\text{p1} = v_1, \text{p2} = i, \text{p3 in bounds} = v_3) \mid i \in \{0, \cdots, 9\}\} & \text{if } p = 2 \\ \{(\text{p1} = v_1, \text{p2} = v_2, \text{p3 in bounds} = i) \mid i \in \{0, 1\}\} & \text{if } p = 3 \end{cases}$$

$$\hat{f}_{\text{SWAP}}((1, 2), (\text{p1} = v_1, \text{p2} = v_2, \text{p3 in bounds} = v_3)) = \{(\text{p1} = v_2, \text{p2} = v_1, \text{p3 in bounds} = v_3)\}$$

Intuitively, $\hat{f}_{\text{PTR}}((dir, p), o)$ produces a set of observations where the dimension corresponding to $p$ is allowed to vary arbitrarily from $o$, and $\hat{f}_{\text{SWAP}}$ swaps the observed values for the given pointers.

### 3.3 CREATING THE TRAINING SET

Our method (Algorithm 1 in the appendix) outputs trees corresponding to elements of $F \times A$, which we label $\text{Oracle}_{f,a}$. Each node (except the root) of $\text{Oracle}_{f,a}$ corresponds to an element in $(F \cup P) \times A \times \{0, 1\}$, and each edge corresponds to an element in $\mathcal{O}$. $\text{Oracle}(f, a, [o_1, \cdots, o_n])$ can be computed by starting at the root of $\text{Oracle}_{f,a}$ and traversing the edges for $o_1, \cdots, o_n$ in turn. Each element in $\bigcup_i Q_i$ (from Section 3.1) maps to a similar traversal of $\text{Oracle}_{f,a}$. Figure 2a shows a partial example for bubblesort.

For training the NPI core LSTM, we extract sequences of the form $((o_1, f_1, a_1, r_1), \cdots, (o_n, f_n, a_n, r_n))$ by performing root-to-leaf traversals on each of $\texttt{Oracle}_{f,a}$; $o_i$ come from the edges, and $f_i, a_i, r_i$ from the nodes. If the NPI core gets 100% accuracy on these sequences, then it is guaranteed to match the oracle's behavior in any setting.

However, due to the approximations made in considering observations instead of environments, some of these sequences may never arise during execution of the oracle on a concrete problem. For example, consider $\texttt{PTR 1 RIGHT}$ in $\texttt{RSHIFT}$ of Figure 1. Before, $\texttt{p1}$ points to one location left of $\texttt{p2}$; after, they point to the same location, so the value for $\texttt{p1}$ and $\texttt{p2}$ must match in the observation. $\hat{f}_{\texttt{PTR}}$ cannot account for this as it is unaware of the pointer locations. Nevertheless, *every* sequence of observations that the oracle may produce from its operations will be present in the set.

## 4    DETECTING AND REMOVING IRRELEVANT OBSERVATIONS

In the NPI architecture, the NPI core receives an observation $o \in \mathcal{O}$ at each step of execution, and the set $\mathcal{O}$ from which the observation is drawn is identical across all functions and all steps. In theory, it is possible to take a different action for each of the possible sequences of observations up to that point in the execution of the function. However, practical NPI functions typically have simple behavior, with many parts of the observation sequence irrelevant for execution and therefore unneeded. For example, the $\texttt{LSHIFT}$ and $\texttt{RSHIFT}$ functions in bubblesort should always execute the same sequence of actions no matter which observation sequence is given.

Therefore, we propose to instead provide the NPI core with observations $\tilde{o} \in \tilde{\mathcal{O}}_{f,a}^{(c_1, \cdots, c_n)}$, where $\tilde{\mathcal{O}}_{f,a}^{(c_1, \cdots, c_n)}$ is a family of sets indexed by a function $f$, argument $a$, and the sequence of actions $c_i$ taken so far ($c_i \in (F \cup P) \times A \times \{0, 1\}$). There exists a function $\mu_{f,a}^{(c_1, \cdots, c_n)} : \mathcal{O} \mapsto \tilde{\mathcal{O}}_{f,a}^{(c_1, \cdots, c_n)}$ for each set in the family; in other words, every element in $\mathcal{O}$ maps to an element in $\tilde{\mathcal{O}}_{f,a}^{(c_1, \cdots, c_n)}$, but the mapping is many-to-one.

At the beginning of executing function $f$ with argument $a$, we obtain observation $o_1$. We will then compute $\mu_{f,a}^{()}(o_1) = \tilde{o}_1$, and provide $\tilde{o}_1$ to the NPI core, producing $c_1$ as the first action. After $c_1$ completes, we obtain the next observation $o_2$, compute $\mu_{f,a}^{(c_1)}(o_2) = \tilde{o}_2$, provide it to the NPI core, and so on. Even though $\mu_{f,a}^{(c_1, \cdots, c_n)}$ is a many-to-one mapping, $\tilde{o}_1, \cdots, \tilde{o}_n$ should contain the salient information from $o_1, \cdots, o_n$ necessary to exactly specify the next action $c_{n+1}$.

By performing this transformation, training and verifying the NPI core requires much less data and computation, since there are not as many behaviors that it needs to learn. This reduction is particularly beneficial for the automatically generated training sets of Section 3.3, because it mostly removes the extraneous execution traces they contain. We also obtain a more parsimonious explanation for the behavior of the oracle.

**Multi-dimensional observations.**    In the tasks and oracles considered by Cai et al. (2017) (including the two algorithmic tasks from Reed & de Freitas (2016)), the observation exposed through the domain-specific encoder has a natural multi-dimensional structure. For example, in the bubblesort task, an observation consists of three dimensions: two digits (value at $\texttt{p1}$ and $\texttt{p2}$) and a boolean value (whether $\texttt{p3}$ is within bounds).

This provides a natural method for constructing $\tilde{\mathcal{O}}_{f,a}^{(c_1, \cdots, c_n)}$: if $\mathcal{O} = \mathcal{X} \times \mathcal{Y} \times Z$, then we can exclude some of the dimensions to form $\tilde{\mathcal{O}}_{f,a}^{(c_1, \cdots, c_n)} = \mathcal{X} \times \mathcal{Y}$ or $\tilde{\mathcal{O}}_{f,a}^{(c_1, \cdots, c_n)} = \mathcal{Z}$ for instance. We can also exclude all dimensions, in which case $\tilde{\mathcal{O}}_{f,a}^{(c_1, \cdots, c_n)}$ would be a singleton set (a nullary Cartesian product). $\mu_{f,a}^{(c_1, \cdots, c_n)}(o)$ simply drops the excluded dimensions of $o$.

**Detecting irrelevant observation dimensions.**    Section 3.3 describes trees where the nodes correspond to NPI actions ($(F \cup P) \times A \times \{0, 1\}$) and the edges correspond to observations, to describe the behavior of the oracle. To determine which observation dimensions are irrelevant and can be excluded in $\tilde{\mathcal{O}}_{f,a}^{(c_1, \cdots, c_n)}$, for each $(f, a)$, we build a tree with actions as edges and nodes as *branching*

*points* which describe the set of observations dimensions necessary to determine the branch to take. Example trees for `BSTEP` and `COMPSWAP` are illustrated in Figure 2b.

We construct these trees from a complete training or verification set which describes all of the possible behaviors of the oracle on any input. We can use the automated method described in Section 3 to obtain this set. Each execution trace of a function $f$ with argument $a$ is a sequence of the form $(o_1, f_1, a_1, r_1), \cdots, (o_n, f_n, a_n, r_n)$. In the tree for $(f, a)$, we ensure that a path exists from the root to a leaf where the $i^{\text{th}}$ edge along this path is labeled with $(f_i, a_i, r_i)$ (without any duplicates, i.e., each node does not have more than one outgoing edge with the same label). Each edge also contains a set of observations, and we add each observation from the execution trace to the set in the corresponding edge. For example, when processing the above execution trace, we add $o_1$ to the edge for $(f_1, a_1, r_1)$ at depth 1; we traverse the edge to reach $n_1$, and add $o_2$ to the outgoing edge of $n_1$ corresponding to $(f_2, a_2, r_2)$, and so on.

Afterwards, we examine each node with more than one outgoing edge, and then decide which subset of observation dimensions would have been sufficient to decide which of the branches to take. For example, in Figure 2b's tree for $(\texttt{BSTEP}, ())$, the root node has two children: for $(\texttt{NOP}, (), 1)$ and $(\texttt{COMPSWAP}, (), 0)$. By looking at the values of $o$ attached to each edge, we can determine that only `p1` is relevant for deciding between the two.

**Using information about irrelevant observation dimensions.** To obtain $\mu_{f,a}^{(c_1, \cdots, c_n)}$, we start at the root of the tree for $f, a$ and traverse the edges labeled with $c_1, \cdots, c_n$. We apply $\mu$ to replace all $o$ with $\tilde{o}$ in our training set of execution traces. This replacement typically results in many redundant traces, and removing them significantly shrinks the set's size. We can then exclusively use $\tilde{o}$ for training and evaluating the neural network.

## 5    EXPERIMENTAL RESULTS

We re-implemented three tasks from Cai et al. (2017): addition, bubblesort, and topological sort. We defined our oracles exactly as described in their paper, automatically generated suitable training sets by making queries to the oracle per Section 3, and minimized them following Section 4.

### 5.1    ARCHITECTURAL DETAILS

For all experiments, we used a 2-layer LSTM with 256 units in each layer as the NPI core. $a_t$ has 64 dimensions, $p_t$ has 256 dimensions, and $k_t$ has 32 dimensions. The LSTM accepts each timestep's input through a 3-layer MLP, where the 1st layer receives $f_{enc}$ and $a_t$ as input, and the second layer additionally receives $p_t$ (concatenated with the output of the first layer).

### 5.2    EMPIRICAL RESULTS OF USING ABSTRACT INTERPRETER TRAINING SET

On all three of the listed tasks, we trained the NPI model with the automatically generated then minimized training sets. We continue training each model until it achieves 100% accuracy on the training set. The automatically generated training sets are also *verification sets* (as defined in Cai et al. (2017)),[2] and so 100% accuracy on the training set indicates that the neural network has learned to copy the oracle perfectly.

The resulting models show empirically perfect generalization. Specifically:

- Addition: 100% accuracy on 80 random problems consisting of 2, 4, 8, 16 digits.
- Bubblesort: 100% accuracy on 100 random arrays of length 2, 4, 8, 20, 50.
- Topological sort: 100% accuracy on 100 random graphs of 5, 6, 7, 8, 70 nodes.

Furthermore, we obtain 100% accuracy on conventional verification sets constructed through manual analysis of the oracle, following the methodology described in Cai et al. (2017).

---

[2] However, unlike Cai et al. (2017), our training set directly contain execution traces for each non-primitive function (such as `BSTEP` and `COMPSWAP` in bubblesort); we do not identify each execution trace as created for a particular input problem (in bubblesort, a concrete array to sort).

Without minimization, we reached 100% training accuracy on the automatically generated training set for topological sort, but not for bubblesort and addition. As explained in Section 3.3, the approximations made due to our method means that many of the generated traces for each function would never occur while running the oracle on an input problem. Different hyperparameters may enable leaning of the un-minimized set, but we did not investigate further.

### 5.3 EFFECT OF REMOVING IRRELEVANT OBSERVATIONS

Removing irrelevant observations can significantly reduce the size of the automatically generated training set.

- Addition: originally 10129737 unique traces in the automatically generated training set, reduced to 704 traces.
- Bubblesort: originally 325622 unique traces, reduced to 137 traces.
- Topological sort: originally 831 unique traces, reduced to 16 traces.

As a point of comparison, we also tried randomly sampling the same number of traces as would be chosen by the minimization, and training the neural network on those training datasets. As expected, these models do not succeed at solving any of the test problems. As such, they also fail to achieve full accuracy on the verification sets.

### 5.4 COMPARISON AGAINST PREVIOUS WORK

We also generated training sets for the three problems, using methodology similar to that used by previous work. Specifically, we consider the systematic approach of generating all problems containing a certain number of digits/elements/nodes, and also randomly generating problems of a certain size. We also generated conventional verification sets using the methodology described by Cai et al. (2017).

The appendix includes the detailed results. To summarize them here: we often fail to learn a verified NPI neural program from many of the training sets, and it is tricky to figure out what a training set should contain to ensure success.

The procedure of Section 4 for removing irrelevant observations allows us to learn correct programs with smaller training sets, by excluding many possible spurious behaviors of the oracle and therefore simplifying learning. However, detecting irrelevant observation dimensions requires a dataset which exhibits all of the possible behaviors of the oracle (such as one generated per Section 3).

In the previous work of Cai et al. (2017), the creation of a verification set was a manual process based on a careful analysis of the oracle. Our work allows us to create a training and verification set automatically, with only black-box queries to the oracle and without any manual analysis needed.

## 6 RELATED WORK

**Active learning.** In semi-supervised learning, we have a set of examples where some are labeled and others are not, and we would like to learn a function from the example to the label. Active learning extends this setting by allowing the system to query the data provider for labels of some of the unlabeled points (Settles, 2010). The goal, then, is to learn the best possible classifier while minimizing the number of queries which need to be made.

Our work is similar in that we assume the existence of an oracle which can provide the correct answer for any example. However, unlike most past work in active learning, our querying does not depend on the status of a machine learning model under training. By assuming a structured oracle which can provide execution traces, we also do not explicitly work in the space of input examples, but rather in terms of the observations which (by assumption) the oracle uses to make its decisions.

**Software testing.** Given a piece of software, we would like to characterize its behavior as completely as possible to ensure that it will not misbehave under any input. In the field of software engineering and programming languages research, many approaches have been developed towards

achieving this goal. Techniques such as *symbolic execution* (King, 1976), *concolic testing* (Godefroid et al., 2005), and *model checking* (Clarke et al., 1999) try to uncover all of the states and behaviors exhibited by a program for the purpose of discovering bugs and security vulnerabilities.

In our work, we also seek to comprehensively describe all of the possible behaviors of an NPI oracle. However, we only assume *black-box* access to the oracle, unlike many software testing techniques which make direct use of the code of the target program. Network protocol inference and fuzz testing are some software testing applications with similar assumptions.

## 7 CONCLUSION

Generalization to complex inputs and inability to provide a proof of correctness have been two challenges faced by most previous work in the space of learning algorithmic tasks with neural networks. While previous work by Cai et al. (2017) provides an approach to address these challenges in the Neural Programmer-Interpreter framework, it assumes the existence of training sets manually designed to contain sufficient diversity and complexity in order to fully describe the task. In this work, we showed how to entirely automate the process of learning NPI programs with provably perfect generalization, through automatic generation of the necessary training and verification sets for the Neural Programmer-Interpreter with only black-box access to an oracle. Furthermore, we discuss how to detect and remove irrelevant observations from execution traces which enables faster and easier learning of the oracle's behavior.

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

---

**Algorithm 1** Iterative algorithm for computing $\texttt{Oracle}_{f,a}$

---

$queue \leftarrow [\forall o \in \text{initial observations} : ((o), initialFunction, initialArg)]$
$nextQueue \leftarrow Queue()$ ▷ $queue$ and $nextQueue$ allow each element to be added only once
$callers \leftarrow \{\}$ ▷ Mapping from $F \times A \times O$ to $2^{F \times A \times [O]}$
$obsMap \leftarrow \{\}$ ▷ Mapping from $F \times A \times \mathcal{O}$ to $2^{\mathcal{O}}$
$oracleTree \leftarrow TreeNode()$
**for all** $f, a \in F \times A$ **do**
    $oracleTree.addChild((f,a), \emptyset)$
**end for**
**repeat**
    **while** $queue \neq []$ **do**
        $(o_1, \cdots, o_n), f, a \leftarrow queue.popLeft()$
        $node \leftarrow oracleTree.traverse((f,a), o_1, \cdots, o_{n-1})$
        $g, a', toReturn \leftarrow \texttt{Oracle}(f, a, [o_1, \cdots, o_n])$ ▷ $g \in F \cup P; a' \in A$
        $node.addChild(o_n, (g, a', toReturn))$ ▷ Record the oracle's response to $(f,a), o_1, \cdots, o_n$.
        **if** $g$ is a primitive function **then**
            $nextObs \leftarrow \hat{g}(a', o_n)$
        **else if** $g$ is a non-primitive function **then**
            $nextObs \leftarrow obsMap(g, a', o_n)$
            $nextQueue.enqueue(((o_n), g, a'))$
            $callers(g, a', o_n).add((f, a, (o_1, \cdots, o_n)))$
        **end if**
        **for all** $\hat{o} \in nextObs$ **do**
            $queue.enqueue(((o_1, \cdots, o_n, \hat{o}), f, a))$
        **end for**
        **if** $toReturn$ **then**
            **for all** $\hat{o} \in nextObs$ **do**
                $obsMap(f, a, o_1).add(\hat{o})$ ▷ Function $f$, run with argument $a$ and initial observation $o_1$, can finish with observation $\hat{o}$.
            **end for**
            **for all** $\hat{f}, \hat{a}, (\hat{o}_1, \cdots, \hat{o}_n) \in callers(f, a, o_1)$ **do**
                $queue.enqueue(((\hat{o}_1, \cdots, \hat{o}_n), \hat{f}, \hat{a}))$ ▷ Re-analyze all callers of $f, a, o_1$.
            **end for**
        **end if**
    **end while**
    $queue \leftarrow nextQueue; nextQueue \leftarrow Queue()$
**until** no changes made to $obsMap$ and $oracleTree$
**return** $obsMap, oracleTree$

---

## A  FULL ALGORITHM USED IN SECTION 3

Algorithm 1 describes the method of Section 3 in detail. We use the following notation:

- $oracleTree$ is a tree where the subtrees rooted at each child of the root correspond to $\texttt{Oracle}_{f,a}$ from Section 3.3. Let us consider the root node to have depth 0. For each element of $(f, a) \in F \times A$, there exists a node at depth 1 (a child node of the root node), and the edge from the root to that node is labeled with $(f, a)$. All nodes at depth 2 or below are labeled with an element of $(F \cup P) \times A \times \{0, 1\}$. All edges from a node at depth $d$ to a child node at depth $d + 1$, for $d \geq 1$, are labeled with an observation (an element from $\mathcal{O}$). For a given node of depth 1 or greater, its outgoing edge labels are unique, and so it has at most $|\mathcal{O}|$ outgoing edges.

- $obsMap : F \times A \times \mathcal{O} \to 2^{\mathcal{O}}$ is an equivalent to $\hat{f}$ defined in Section 3.2. However, the domain of $obsMap$ is over non-primitive functions instead of primitive functions.

- $TreeNode$ represents a node in a tree and all of its descendants. Each node and edge has a label. It has the following interface:

- – $node.traverse(x_1, \cdots, x_n)$: Follow the edges labeled with $x_1, \cdots, x_n$ and return the resulting node, which will have depth $n$.
  - – $node.addChild(o, v)$ adds an outgoing edge labeled with $o$, connecting to a (new) node labeled with $v$. It does nothing if there already exists an outgoing edge labeled with $o$.
- • $callers$ and $obsMap$ are multimaps. $callers(k)$ and $obsMap(k)$ returns the set of values with given key from the map. $callers(k).add(...)$ and $obsMap(k).add(...)$ adds a value to this set.
- • $queue.popLeft()$ removes and returns the leftmost element of $queue$. $queue.enqueue(x)$ adds a new element to the right end of $queue$. However, if $x$ has already been added to $queue$, then $enqueue$ does nothing, even if $x$ was already returned by $popLeft()$.

**Theorem.** *Assuming that each function of the oracle always executes for at most $k$ steps for some fixed $k$, and the possible number of observations is finite, Algorithm 1 terminates.*

*Proof.* Consider traversing the path from the root of $oracleTree$ to an internal node, and reading the labels of edges present along the path. We obtain an element of $F \times A \times \mathcal{O}^{\leq k}$, where $F \times A$ comes from the first edge, and $\mathcal{O}^{\leq k}$ comes from all subsequent edges ($\leq k$ because the tree would have at most $k$ depth).

$\xi : oracleTree \to 2^{F \times A \times \mathcal{O}^{\leq k}}$ performs this traversal over all such paths, and returns their combined result: a set where each element is from $F \times A \times \mathcal{O}^{\leq k}$.

Let us denote the the effect of the outer loop of Algorithm 1 on $oracleTree$ as $x_{i+1} = f(x_i)$, where $x_i$ is the old value of $oracleTree$ and $x_{i+1}$ is the new value. $\xi(x) \subseteq \xi(f(x))$, because $f$ can only add new nodes and edges to $oracleTree$, and does not delete nodes or change their labels.

We can consider a partial ordering over $2^{F \times A \times \mathcal{O}^{\leq k}}$. Then $\xi(x) \leq \xi(f(x))$, and $2^{F \times A \times \mathcal{O}^{\leq k}}$ has a maximal element with respect to this partial ordering (the set of all elements of $F \times A \times \mathcal{O}^{\leq k}$, so $x$ cannot grow indefinitely.

Similarly, we can treat possible states of $obsMap$ as elements of $2^{F \times A \times \mathcal{O} \times 2^{\mathcal{O}}}$. Like $oracleTree$, the loop in Algorithm 1 only adds new elements to $obsMap$. Following the same argument as in the previous paragraph, $obsMap$ cannot grow indefinitely either.

When $oracleTree$ and $obsMap$ eventually stops growing, the outer loop in Algorithm 1 will terminate, as specified in the pseudocode.

$\square$

**Theorem.** *When Algorithm 1 terminates,* oracleTree *contains all possible behaviors of the oracle, including all sequences of observations and actions which might occur when the oracle runs on any valid input problem.*

*Proof.* We will first prove the theorem for a less efficient version of Algorithm 1, where each item removed from $queue$ is also added to $nextQueue$. In this version, there exists a corresponding item in $queue$ or $nextQueue$ for each node in $oracleTree$, because each item removed from $queue$ creates at most one new node in $oracleTree$. By the previous theorem, there exists an iteration where the algorithm terminates because no changes have been made to $obsMap$ and $oracleTree$. In this iteration, there exists an item in $queue$ for each node in $oracleTree$, and we want to show that $oracleTree$ already contains all possible behaviors of the oracle (since we will not be modifying $oracleTree$ further before returning).

For the purposes of the proof, we will assign a *generation* to each entry in each set contained within $obsMap$. Since $obsMap$ is finite, the number of generations is also finite. To reiterate, $obsMap(f, a, o) \in 2^{\mathcal{O}}$ gives, when executing non-primitive function $f$ with argument $a$ and initial observation $o$, the set of resulting observations after $f$ returns. If $obsMap(f, a, o)$ is empty, it means that execution of $(f, a, o)$ never terminates, no matter which observation is selected at each node in the execution tree (we assume that our oracles are well-behaved, and has no such functions).

At the start of the algorithm, we initialize $queue$ with tuples containing $initialFunction$, $initialArgument$, and observations in the set of initial observations. We assign generation 0

to the final observations in $obsMap$ produced by execution traces starting at some $(f, a, o)$ that (1) are reachable from the initial $queue$ without using $obsMap$, and (2) contain only calls to primitive functions. To clarify, satisfying (1) means (i) $(f, a, o)$ either needs to be in the initial $queue$; or (ii) invoked within an execution trace of a function in the initial $queue$ before any non-primitive functions have been called in that trace; (iii) invoked within an execution trace of a function satisfying (ii) before any non-primitive functions have been called in that trace; and so on recursively.

Generation $n$ is assigned to the final observations produced by execution traces starting at some $(f, a, o)$ that (1) are reachable from the initial queue using only entries in $obsMap$ belonging to generations $i < n$, and (2) when encountering a node corresponding to a non-primitive function call while traversing the execution tree, chose an outgoing edge for an observation corresponding to generation $i$ (for both, $i < n$; we choose the the smallest possible $n$).

In the final idempotent iteration of the modified version of Algorithm 1, we will consider every node inside $oracleTree$, but make no changes to $obsMap$ or $oracleTree$. Using induction on $n$, we will now show that at the start of this iteration, $obsMap(f, a, o)$ is an overcomplete approximation to the true behavior of the oracle.

- *At termination, all items which should belong to generation 0 are present in $obsMap$.*

  There are two parts to check. First, $queue$ contains each $(f, a, o)$ that has a final observation in generation 0; second, for those $(f, a, o)$, $obsMap(f, a, o)$ actually contains those generation 0 final observations.

  To check the first part: each $(f, a, o)$ for generation 0 is either in the initial $queue$, in which case it should still be in $queue$; otherwise, we can see it would have been added to $nextQueue$ through some path of function invocations starting at an initial entry of $queue$.

  For the second part, we assumed that $\hat{g}$ specifies the behavior of primitive functions in an overcomplete way. Since the final iteration is idempotent by assumption, we can be confident we have already explored all the possible execution traces consisting only of primitive function calls, and therefore the corresponding entries in $obsMap$ are also present.

- *Assuming generations $0, \cdots, n-1$ are present in $obsMap$, generation $n$ is also present.*

  The argument is similar to the base case, except we are now allowed to use parts of $obsMap$ which we have shown are present. For the first part of the argument, those parts of $obsMap$ allow us to reach more $(f, a, o)$ tuples that are only called from execution traces containing more than one non-primitive function call. For the second part, we use not only $\hat{g}$ but also the parts of $obsMap$ we have already assumed presence.

Now we have shown that $obsMap$ is complete, it is straightforward to see $oracleTree$ is also complete in this iteration; as if it were not, we would necessarily end up modifying $oracleTree$ to add something from $obsMap$ (or $\hat{g}$).

We can also see that we do not need to make the modification to Algorithm 1 assumed at the beginning of the proof in order for it to be correct. After an entry has appeared on and been processed from $queue$, it is only relevant for that entry to be placed upon $queue$ again if its renewed presence on it will lead to some change in $oracleTree$. This only happens when $obsMap$ changes, and indeed we keep track of $callers$ to re-add items to $queue$ at that time.

$\square$

## A.1 REQUIREMENTS AND ASSUMPTIONS

We list the requirements on the oracle used throughout the paper. The oracles for the tasks evaluated in our paper (addition, bubblesort, and topological sort from Cai et al. (2017)) meet all of the requirements.

**Oracle is deterministic.** We assume that `Oracle` will always give the same output on the same input.

| | No. of function traces | | Success with | |
|---|---|---|---|---|
| | Orig. | Mini. | Orig. | Mini. |
| Systematic, 2 digits; 9900 problems | 108900 | 647 | N | N |
| Systematic, 2 + 3 digits; 162000 problems | 2145240 | 692 | N | N |
| Random, 4 digits; 10000 problems | 149656 | 704 | N | Y |
| Random, 4 digits; 200000 problems | 2994804 | 704 | Y | Y |

Table 2: Conventional training sets for addition.

| | No. of function traces | | Success with | |
|---|---|---|---|---|
| | Orig. | Mini. | Orig. | Mini. |
| Systematic, length 2; 100 problems | 2100 | 127 | Y | Y |
| Systematic, length 2 + 3; 1100 problems | 45100 | 127 | Y | Y |
| Systematic, length 2 + 3 + 4; 11100 problems | 775100 | 127 | Y | Y |
| Random, length 5; 100 problems | 11100 | 126 | Y | Y |
| Random, length 5; 1100 problems | 122100 | 127 | Y | Y |
| Random, length 5; 11100 problems | 1232100 | 127 | N | Y |

Table 3: Conventional training sets for bubblesort.

$\mathcal{O}$, **the set of observations, is finite.**   We need to be able to enumerate the set of possible observations. This is not a hurdle for the computational problems as considered in Cai et al. (2017), but poses a challenge for tasks like 3D model canonicalization from Reed & de Freitas (2016), where the observation was a bitmap image.

**Primitive functions are deterministic, and we can compute $\hat{f}$.**   A primitive function $f : A \times E \to E$, operates on some environment $e_i$ and transforms it into a different environment $e_j$.

In Section 3.2, we defined the corresponding $\hat{f} : A \times \mathcal{O} \to 2^{\mathcal{O}}$, which maps from observations to sets of observations.

In order to use our method, we must know enough about the environment and defined primitive functions so that we can efficiently compute $\hat{f}$ for each primitive function $f$.

**Set of initial observations is known.**   An oracle which can solve a problem like addition or sorting will require that the input be encoded in the environment in a particular way. If the domain of inputs is known, it is straightforward to then determine the set of possible observations that would arise once we have encoded each possible input into the environment as demanded.

**Each function always executes for a bounded number of steps.**   Assuming only black-box access to the oracle, describing all of the possible behaviors of a function finitely is impossible unless each function only executes for a bounded number of steps. The oracles considered in Cai et al. (2017) satisfy this assumption, but the ones in Reed & de Freitas (2016) do not, as those contain functions which execute for a variable number of steps depending on the size of the input.

**Oracle terminates on at least one input.**   We assume that the oracle will complete on at least one problem within a bounded number of steps. Most useful oracles will terminate in finite time for any input, not just at least one.

## B   RESULTS WITH CONVENTIONAL TRAINING SETS

Tables 2, 3, 4 show results when we train the NPI model on conventionally created training sets. In these tables, 'mini.'' denotes application of the procedure in Section 4.

| | No. of function traces | | Success with | |
|---|---|---|---|---|
| | Orig. | Mini. | Orig. | Mini. |
| Systematic, 2 + 3 nodes; 10 problems | 174 | 14 | N | Y |
| Systematic, 2 + 3 + 4 nodes; 74 problems | 1846 | 14 | Y | Y |
| Systematic, 2 + 3 + 4 + 5 nodes; 1098 problems | 36726 | 14 | Y | Y |
| Random, 5 nodes; 10 problems | 341 | 14 | N | Y |
| Random, 5 nodes; 100 problems | 3413 | 14 | Y | Y |
| Random, 5 nodes; 1000 problems | 34078 | 14 | Y | Y |

Table 4: Conventional training sets for topological sort.

To determine whether we could learn a correct NPI program with a given dataset, we ran the training procedure for at least as many iterations as we needed to obtain a verified model when training on any other dataset.

We generated problems systematically in the following way:

- Addition: all pairs of numbers where at least one number is 2 digits long (with no leading 0s), or all pairs of numbers where one number is 2 digits long and the other is 3 digits long.

- Bubblesort: all arrays of given length, where each element in the array can be one of the digits between 0 and 9.

- Topological sort: all DAGs with a given number of nodes. We number all nodes, and generate a complete DAG by adding an edge from node $i$ to $j$ if $i < j$. We consider all DAGs created by removing some set of edges from this complete DAG.

These tables show that it can be difficult to figure out what demonstrations need to be provided in the training set in order to learn the correct NPI program. To generate a suitable training set without trial and error, the creator needs to have a full understanding of how the oracle functions internally, which is counter to the original goal of automatically learning the behavior of a given program. The methods in our paper can fully automate this process.

## C   SAMPLE COMPLEXITY OF PAST WORK IN NEURAL PROGRAM LEARNING

| Paper | Tasks | Training data |
|---|---|---|
| Grefenstette et al. (2015) | Sequence copying, sequence reversal | Dynamic |
| Joulin & Mikolov (2015a) | Counting, memorization, binary addition | Dynamic |
| Zaremba & Sutskever (2015) | Repeated copy, sequence reversal | Dynamic |
| Kaiser & Sutskever (2016) | Binary addition, binary multiplication, copying, reversing, duplicating, counting | 200,000 |
| Reed & de Freitas (2016) | Decimal addition, bubblesort | 640/1216 |
| Kurach et al. (2016) | BST traversal, array merge, linked list search, etc. | Dynamic |
| Zaremba et al. (2016) | Copy, reverse, walk, addition, multiplication | Dynamic |
| Graves et al. (2016) | Graph traversal, shortest path, logical inference, mini-SHRDLU | Dynamic |
| Price et al. (2016) | Multiplication | Dynamic |
| Freivalds & Liepins (2017) | Multiplication | 200,000 |

In the above table, we summarize the number of training examples that past works in neural program learning reported using for their experiments. "Dynamic" indicates that, to the best of our knowledge from reading the papers and also any available code, the training data used for each step of training was generated randomly on-the-fly. In other words, the training data is equivalent to the set of *all* possible problems of a given complexity, and the training procedure samples from this set with replacement. Often, the complexity of the problems in each mini-batch will be adjusted dynamically through the use of curriculum learning.