# OpenReview forum: "Learning what to learn in a neural program"
_ICLR.cc/2018/Conference — Reject_

### Official Review · AnonReviewer3 · 2017-11-25
**Improves data efficiency of NPI, but makes stronger assumptions.**

**Rating:** 5
**Confidence:** 4

**Review:**

Quality
The paper is well-written and clear, and includes relevant comparisons to previous work (NPI and recursive NPI).

Clarity
The paper is clearly written.

Originality
To my knowledge the method proposed in this work is novel. It is the first to study constructing minimal training sets for NPI given a black-box oracle. However, as pointed out by the authors, there is a lot of similar prior work in software testing.

Significance
The work could be potentially significant, but there are some very strong assumptions made in the paper that could limit the impact. If the NPI has access to a black-box oracle, it is not clear what is the use of training an NPI in the first place. It would be very helpful to describe a potential scenario where the proposed approach could be useful. Also, it is assumed that the number of possible inputs is finite (also true for the recursive NPI paper), and it is not clear what techniques or lessons of this paper might transfer to tasks with perceptual inputs. The main technical contribution is the search procedure to find minimal training sets and pare down the observation size, and the empirical validation of the idea on several algorithmic tasks.

Pros
- Greatly improves the data efficiency of recursive NPI.
- Training and verification sets are automatically generated by the proposed method.

Cons
- Requires access to a black-box oracle to construct the dataset.
- Not clear that the idea will be useful in more complex domains with unbounded inputs.

---

> ### Author Response · Authors · 2017-12-22
> **Reply to review**
>
> Thanks for your thoughtful comments!
>
> Regarding the motivation for training an NPI when we have a black-box oracle:
> To our knowledge, prior work in learning algorithms with neural networks has largely assumed access to an executable oracle. We have added a section in the appendix summarizing how many training examples were used by past work in program learning; the vast majority of them randomly generate fresh problem instances at each training step, and then use an oracle to get the solutions. Indeed, as did many of these prior works, we used an executable oracle for the experiments in our work. However, the methods in our work do not require that the oracle be an executable computer program; it can be any source which provides the relevant demonstrations, such as a human. Furthermore, after we have learned an NPI program, we no longer need any access to the oracle in order to perform the same function as the oracle. This is highly useful if it is expensive to obtain responses from the oracle.
>
> Regarding perceptual inputs:
> We agree that tasks with perceptual inputs are an important domain, and that it is difficult to apply techniques from this paper to such tasks. However, a central focus of this work is to be able to provide a complete and formal proof of the learned NPI program's correctness, so that we can be sure it is equivalent to an oracle in every relevant way. If the set of possible inputs is effectively infinite, it is not really feasible to test a black box oracle's response to all of them in order to be able to replicate the oracle exactly. We anticipate that it will be necessary to make a very different set of assumptions in order to provide similar guarantees for tasks using perceptual inputs, and that the nature of the guarantees will also be different.
>
> For example, consider the following approach to working with perceptual inputs. Assuming that the oracle only relies upon some aspect of each perceptual input that lies within a finite space, we may be able to decompose the input encoder into two parts: one which extracts that aspect of the input, and the other which directly encodes the aspect for the core. If we were sorting or adding numbers represented as MNIST digits, we could use a digit classifier and then provide the output of that classifier to NPI. If we take the digit classifier as externally provided and assume that it is correct, we could proceed with techniques used in this paper.
>
> Alternatively, we can envision a class of approaches where we train the perceptual input encoder end-to-end on execution traces to only encode aspects of the input that are salient for reproducing the traces. While such approaches may lead to various interesting results, it will be hard to formally show that the trained model exactly matches the oracle in all situations.
>
> To summarize, the techniques in the paper are geared towards thoroughly addressing a class of tasks that have been considered in many past papers. We leave methods for perceptual inputs to future work, especially given that such work will need different assumptions and lead to a qualitatively different result.

---

### Official Review · AnonReviewer2 · 2017-11-27
**Motivation**

**Rating:** 4
**Confidence:** 4

**Review:**

In this paper, the authors consider the problem of generating a training data set for the neural programmer-interpreter from an executable oracle. In particular, they aim at generating a complete set that fully specifies the behavior of the oracle. The authors propose a technique that achieves this aim by borrowing ideas from programming language and abstract interpretation. The technique systematically interacts with the oracle using observations, which are abstractions of environment states, and it is guaranteed to produce a data set that completely specifies the oracle. The authors later describes how to improve this technique by further equating certain observations and exploring only one in each equivalence class. Their experiments show that this improve technique can produce complete training sets for three programs.

It is nice to see the application of ideas from different areas for learning-related questions. However, there is one thing that bothers me again and again. Why do we need a data-generation technique in the paper at all? Typically, we are given a set of data, not an oracle that can generate such data, and our task is to learn something from the data. If we have an executable oracle, it is now clear to me why we want to replicate this oracle by an instance of the neural programmer-interpreter. One thing that I can see is that the technique in the paper can be used when we do research on the neural programmer-interpreter. During research, we have multiple executable oracles and need to produce good training data from them. The authors' technique may let us do this data-generation easily. But this benefit to the researchers does not seem to be strong enough for the acceptance at ICLR'18.

---

> ### Author Response · Authors · 2017-12-22
> **Reply to review**
>
> Thank you for your thoughtful comments!
>
> Regarding why we need a data generation technique:
> The setting in this paper is closer to active learning, where we assume that we can query an oracle with previously unlabeled data points to obtain more labels. Our goal is to learn the true underlying program. However, if we are only given a fixed set of data, it could easily be that this data does not demonstrate all of the behaviors of the latent program. As an example, tables 2, 3, and 4 in the appendix demonstrate that when training the NPI architecture is trained on various manually constructed data sets, the resulting model can fail to generalize.
>
> In the general case, it may be infeasible to devise a set of queries to an oracle such that we can exactly learn the underlying rule being employed by the oracle. However, for our paper, we were able to build upon a formulation of the program-learning problem from prior work (most importantly, recursive NPI from Cai et al). By making use of the underlying structure provided by recursive NPI, we show how to create a dataset that demonstrates all of the possible behaviors of the oracle. Using this dataset, we can obtain a trained NPI program which exhibits perfect generalization, and formally prove its generalization ability.
>
> Regarding why we would like to replicate an executable oracle:
> To our knowledge, prior work in learning algorithms with neural networks has largely assumed access to an executable oracle. We have added a section in the appendix summarizing how many training examples were used by past work in program learning; the vast majority of them randomly generate fresh problem instances at each training step, and then use an oracle to get the solutions. Indeed, as did many of these prior works, we used an executable oracle for the experiments in our work. However, the methods in our work do not require that the oracle be an executable computer program; it can be any source which provides the relevant demonstrations, such as a human. Furthermore, after we have learned an NPI program, we no longer need any access to the oracle in order to perform the same function as the oracle. This is highly useful if it is expensive to obtain responses from the oracle.

---

### Official Review · AnonReviewer1 · 2017-12-05
**Issue with scalability makes me not like the work**

**Rating:** 5
**Confidence:** 2

**Review:**

Previous work by Cai et al. (2017) shows how to use Neural Programmer-Interpreter (NPI) framework to prove correctness of a learned neural network program by introducing recursion. It requires generation of a diverse training set consisting of execution traces which describe in detail the role of each function in solving a given input problem. Moreover, the traces need to be recursive: each function only takes a finite, bounded number of actions. In this paper, the authors show how training set can be generated automatically satisfying the conditions of Cai et al.'s paper. They iteratively explore all
possible behaviors of the oracle in a breadth-first manner, and the bounded nature of the recursive
oracle ensures that the procedure converges. As a running example, they show how this can be be done for bubblesort. The training set generated in this process may have a lot of duplicates, and the authors show how these duplicates can possibly be removed. It indeeds shows a dramatic reduction in the number of training samples for the three experiments that have been shown in the paper.

I am not an expert in this area, so it is difficult for me to judge the technical merit of the work. My feeling from reading the paper is that it is rather incremental over Cai et al. I am impressed by the results of the three experiments that have been shown here, specifically, the reduction in the training samples once they have been generated is significant. But these are also the same set of experiments performed by Cai et al.

Given the original number of traces generated is huge, I do not understand, why this method is at all practical. This also explains why the authors have just tested the performance on extremely small sized data. It will not scale. So, I am hesitant accepting the paper. I would have been more enthusiastic if the authors had proposed a way to combine the training space exploration as well as removing redundant traces together to make the whole process more scalable and done experiments on reasonably sized data.

---

> ### Author Response · Authors · 2017-12-22
> **Reply to review**
>
> Thank you for your thoughtful comments!
>
> Regarding incrementality:
> We evaluate the same tasks as Cai et al. for purposes of comparison, and to show that our methods apply to a setting proposed in existing work; we did not want to create artificially simple programs that are tailored to the assumptions made in our approach.
>
> We would argue that the experimental results are not the main point of the paper; after all, the previous work of Cai et al. already showed empirical perfect generalization. The main contribution of this work is that we no longer need to manually construct a training set that demonstrates the possible behaviors of a program to be learned. As shown in tables 2, 3, and 4 of the appendix, constructing such a training set is tricky; there exists an unknown threshold (depending on the program to be learned) in terms of how many demonstrations are needed, and how diverse they should be, before the model can learn the correct program.
>
> Furthermore, while a central contribution of Cai et al. is to formally prove that a given learned neural program will generalize to any example, the proof still requires substantial manual effort. In this work, we automate this proof of generalization, as the training set constructed by our method fully describes the oracle's behavior and therefore also serves as the verification set which certifies correctness of the learned NPI program.
>
> Regarding size of the data for the experiments:
> We would like to emphasize that the data-generation method (a main contribution of our paper) is independent of the complexity of running the trained NPI program. Once we have trained a NPI program using the dataset generated by our method (i.e. learned the weights for LSTM and the environmental observation encoder), the computational complexity of running the NPI program is not any different from an equivalent NPI program.
>
> This was our guess for what you meant by "the authors have just tested the performance on extremely small sized data", but we were not entirely sure. Could you clarify your comment so that we can see if it's possible to address it more completely?
>
> Regarding combining the training space exploration as well as removing redundant traces:
> Unfortunately, we do not believe it is possible (in general) to combine these two operations together due to the black-box nature of the oracle. If we exclude certain observation dimensions during training space exploration, we are necessarily not querying the oracle with some observation sequences which could arise during an actual execution of the program on that oracle. It could be that these unqueried observation sequences lead to unexpected behavior of the oracle, which we would not learn.

---

### Public Comment · (anonymous) · 2017-11-03
**High-level motivation**

I'm a bit confused at what the role of the learned NPI component in the paper is. The authors describe a method to construct a method to build a set of examples that describes /all/ program behaviours. Then, they train an NPI on this. However, as /all/ behaviours are known already, it should be possible to derive a deterministic implementation (as a lookup table in the samples). What value does training the NPI add?

---

> ### Author Response · Authors · 2017-11-12
> **Reply to question**
>
> Thanks for your question! The high-level motivation for our work follows from the challenges unaddressed by previous work. Reed & de Freitas [1] showed that by providing structured supervision, it is possible to learn compositional models of program behavior. However, the learned programs fail to behave correctly when run on inputs of greater length than used during training. Cai et al. [2] addressed the problem of generalizability by adding recursive structure to the execution traces used as supervision, which ensured that the learned models can generalize to inputs of arbitrary length.
>
> However, these past works did not address the problem of what the training set should contain in order to learn a program successfully. Furthermore, while Cai et al. described how to verify that a learned neural program has perfect generalizability, the procedure described was fully manual. Our work addresses these challenges and fully automate the process of learning a NPI program with perfect generalization for a given task. As such, the training of NPI follows from the context set by the previous work.
>
> [1] Scott Reed and Nando de Freitas. Neural programmer-interpreters. ICLR 2016.
> [2] Jonathon Cai, Richard Shin, and Dawn Song. Making neural programming architectures generalize via recursion. ICLR 2017.

---

> > ### Public Comment · (anonymous) · 2017-11-13
> > **Role of NPI**
> >
> > Thanks for the reply, though my main question remains unanswered. I understand that with your procedure, you can obtain a subset of the set of all traces that fully specifies the program behavior. But then, why does the NPI need to be trained on this? Wouldn't a method that just searches through the minimized trace set to find what the next operation should be work just as well, without requiring the whole RNN infrastructure? [and faster as well, without all the linear algebra...]

---

### Decision · Program_Chairs · 2018-01-29
**ICLR 2018 Conference Acceptance Decision**

**Decision:**

Reject

**Comment:**

This paper is novel, but relatively incremental and relatively niche; the reviewers (despite discussion) are still unsure why this approach is needed.